# Integrated Transcriptome and Proteome Analysis Reveals the Regulatory Mechanism of Root Growth by Protein Disulfide Isomerase in Arabidopsis

**DOI:** 10.3390/ijms25073596

**Published:** 2024-03-22

**Authors:** Yanan Liu, Peng Song, Meilin Yan, Jinmei Luo, Yingjuan Wang, Fenggui Fan

**Affiliations:** State Key Laboratory of Biotechnology of Shannxi Province, College of Life Science, Northwest University, Xi’an 710069, China; yananliu0115@163.com (Y.L.); speng291002@163.com (P.S.); 18792903260@163.com (M.Y.); ljinmei0331@163.com (J.L.)

**Keywords:** protein disulfide isomerase, transcriptome, proteome, root, Arabidopsis, 16F16

## Abstract

Protein disulfide isomerase (PDI, EC 5.3.4.1) is a thiol-disulfide oxidoreductase that plays a crucial role in catalyzing the oxidation and rearrangement of disulfides in substrate proteins. In plants, PDI is primarily involved in regulating seed germination and development, facilitating the oxidative folding of storage proteins in the endosperm, and also contributing to the formation of pollen. However, the role of PDI in root growth has not been previously studied. This research investigated the impact of PDI gene deficiency in plants by using 16F16 [2-(2-Chloroacetyl)-2,3,4,9-tetrahydro-1-methyl-1H-pyrido[3,4-b]indole-1-carboxylic acid methyl ester], a small-molecule inhibitor of PDI, to remove functional redundancy. The results showed that the growth of Arabidopsis roots was significantly inhibited when treated with 16F16. To further investigate the effects of 16F16 treatment, we conducted expression profiling of treated roots using RNA sequencing and a Tandem Mass Tag (TMT)-based quantitative proteomics approach at both the transcriptomic and proteomic levels. Our analysis revealed 994 differentially expressed genes (DEGs) at the transcript level, which were predominantly enriched in pathways associated with “phenylpropane biosynthesis”, “plant hormone signal transduction”, “plant−pathogen interaction” and “starch and sucrose metabolism” pathways. Additionally, we identified 120 differentially expressed proteins (DEPs) at the protein level. These proteins were mainly enriched in pathways such as “phenylpropanoid biosynthesis”, “photosynthesis”, “biosynthesis of various plant secondary metabolites”, and “biosynthesis of secondary metabolites” pathways. The comprehensive transcriptome and proteome analyses revealed a regulatory network for root shortening in Arabidopsis seedlings under 16F16 treatment, mainly involving phenylpropane biosynthesis and plant hormone signal transduction pathways. This study enhances our understanding of the significant role of PDIs in Arabidopsis root growth and provides insights into the regulatory mechanisms of root shortening following 16F16 treatment.

## 1. Introduction

Protein disulfide isomerases (PDI) are thiol disulfide oxidoreductases that play a crucial role in catalyzing the oxidation, reduction, or isomerization of disulfide bonds in newly synthesized secreted proteins and membrane proteins [1]. Additionally, certain PDIs exhibit chaperone activity, aiding other proteins in rectifying folding errors [2]. The structure of PDI comprises two thioredoxin structural domains with catalytic CXXC motifs, labeled a and a’, respectively [1]. They play a key redox catalytic role in catalytic reactions. In addition, PDI contains two inactive structural domains, referred to as b and b’, with a thioredoxin-like folded structure. By facilitating the redox transition of the thioredoxin structural domains, PDI aids in guiding proteins towards the correct conformation during both folding and unfolding processes [3].

The correct formation of disulfides, also known as oxidative protein folding, is essential in the folding of nascent peptides into natural proteins [4]. The process of generating a natural protein conformation involves both disulfide bonds formation and isomerization steps [5]. PDI catalyzes the formation of a sulfur–sulfur bond between two cysteine residues. This process enhances the stability of the protein structure, enabling it to fold accurately into a specific three-dimensional shape [6]. Secretory proteins and cell surface membrane proteins need to undergo folding and modification processes in the endoplasmic reticulum to achieve a mature and stable state [7,8]. Properly formed disulfide bonds play a crucial role in maintaining the correct structure and function of these proteins, as well as aiding in their normal secretion and localization to the cell surface.

PDI in higher plants play crucial roles in signal transduction pathways and transcriptional complexes that govern gene responses to environmental cues. It was shown that PDI is involved in storage protein folding during seed endosperm formation [9,10]; PDIL1-1 controls endosperm development by regulating the amount and components of proteins in rice seeds [11]; AtPDI5 is essential for proper seed development and regulates programmed cell death timing [12]; AtPDIL2–1, also known as AtPDI11, is directly involved in ovule structure and embryo sac development and determining proper direction of pollen tube growth [13]; and most wheat PDI and PDIL genes are expressed during endosperm development, indicating their association with storage protein biosynthesis and deposition [14]. PDI is involved in the adversity response, with AtPDI11 being necessary for plant growth under reducing stress conditions [15]. AtPDI1 can be induced to express by high salt as well as ABA [16]; AtPDI5 interacts with At4a-like1 in drought stress response [17]; and AtPDI9 affects pollen viability and correct pollen exon formation during heat stress [18]. These findings suggest that PDIs may have a significant role in responding to abiotic stresses.

Transcriptomics (RNA-Seq) is a widely utilized and effective molecular biology analytical method. It allows for the identification, quantification, and functional analysis of RNA in cells, tissues, or biological samples, providing valuable insights into gene expression regulation, signaling pathways, and cellular functions. As biological processes are primarily governed by proteins, proteome research offers a glimpse into the organism’s status under particular conditions [19]. Complementing other genomics techniques, such as genomics and transcriptomics, proteomics helps identify an organism’s proteins and understand the structure and function of specific proteins [20]. In recent years, combined transcriptomic and proteomic techniques have been extensively used to study plant growth and responses under stress conditions [21,22,23,24]. These studies lay a foundation for a deeper understanding of plant adaptation mechanisms to environmental changes, as well as crop improvement, stress prevention, and control.

16F16[2-(2-Chloroacetyl)-2,3,4,9-tetrahydro-1-methyl-1H-pyrido[3,4-b]indole-1-carboxylic acid methyl ester] is a small molecule that has the ability to inhibit protein disulfide isomerase [25]. It is demonstrated that the covalent interaction between a small molecule and its target protein can enhance the efficiency of target identification. Compound 16F16, containing a chloroacetyl moiety, was observed to covalently bind to its target protein, resulting in the inhibition of the target protein’s activity [25]. The Arabidopsis PDI family contains 14 family members with functional redundancy [26]. Previously, the investigation of PDI functions has mainly focused on embryos, embryo sac development, and the oxidative folding of storage proteins. However, there is a noticeable gap in the literature regarding the impact of PDI on Arabidopsis root development. To address this gap, we subjected Arabidopsis to stress treatment with 16F16, followed by comprehensive quantification of transcripts and proteins. Using multi-omics analyses as well as genetics, cell biology, and biochemistry experiments, we found that Arabidopsis root growth and development were inhibited with increasing concentrations of 16F16 treatment in a dose-dependent manner. Our study reveals the potential functions and mechanisms of AtPDIs in root growth and development, and lays the foundation for future studies of PDI function in Arabidopsis or other plants.

## 2. Results

### 2.1. 16F16 Inhibits the Activity of AtPDI In Vitro

It was demonstrated that the PDI inhibitor-16F16 covalently modifies free cysteine thiols, leading to the inhibition of bovine and human PDI activity in vitro [25,27]. In Arabidopsis, PDI-L members AtPDI2/5/6 mainly serve as an isomerase, while PDI-M/S members AtPDI9/10/11 are more efficient in accepting oxidizing equivalents from AtERO1 and catalyzing disulfide bond formation [28,29]. So, we selected AtPDI5 and AtPDI9 for experimentation and assessed their impact on AtPDI activity by exposing them to varying concentrations of 16F16 (0.5 μM, 5 μM, and 50 μM). Our findings revealed a gradual decrease in the reductase activity of PDI as the concentration of 16F16 inhibitor increased, with the highest level of inhibition observed at 50 μM. The results demonstrate a positive correlation between the concentration of the inhibitor 16F16 and the inhibition of AtPDI activity (Figure 1A,B). Overall, our results not only validate previous research, but also provide further evidence of the inhibitory effect of 16F16 on AtPDIs.

### 2.2. Effect of 16F16 Stress on Root Elongation in Arabidopsis

To investigate the impact of 16F16 treatment on root growth, a dose–response test for root elongation was conducted. The results showed that treatment with 16F16 had a significant inhibitory effect on root growth of Arabidopsis, with almost no root growth when added 5 μM 16F16 (Figure 2A). Quantitative analysis indicated a decrease in root length of 16F16-treated seedlings by 65.64%, 58.04%, 47.13%, 35.35% and 10.15%, respectively, compared to the control (Figure 2B). These results indicated that 16F16 had a significant inhibitory effect on root growth, and the inhibitory effect on root length was gradually strengthened with the increase of 16F16 concentration.

### 2.3. Transcriptome and Proteome Data Quality Assessment

To examine the impact of the PDI inhibitor 16F16 on the expression of Arabidopsis-related genes in gene regulation and identify differentially expressed genes (DEGs), this study utilized wild-type and PDI inhibitor 16F16-treated Arabidopsis for total RNA extraction and the creation of high-quality cDNA libraries. The percentage of Q30 bases in each sample was above 93.58%, and the percentage of ambiguous bases was below 0.000748%. The results affirm the exceptional quality of the RNA-Seq dataset, satisfying the required standards for further bioinformatics analysis (Table 1). In the principal component analysis (PCA) plot, replicates within each treatment tended to cluster together, with PC1 accounting for 82% of the variance and PC2 accounting for 10% of the variance (Figure 3A). Pearson correlations among all treatment replicates ranged from 0.90 to 0.99 (Figure 3B).

Proteomic data was collected from treated and control groups as test samples, each group having four replicates. The samples underwent Tandem Mass Tag (TMT) analysis, resulting in the identification of 54,762 peptides and 8559 proteins. The distribution of protein molecular weights (MW) was broad, ranging from 1 to 750 kDa. Peptide lengths for each protein showed that the majority (90%) fell within the 7–22 amino acid (aa) range (Figure 3C). In addition, most proteins had a sequence range of less than 30% (Figure 3D). These results demonstrate the reliability of the sequencing results for subsequent data analysis.

### 2.4. Analysis of DEGs and DEPs and Subcellular Localization of DEPs

Following bioinformatics and statistical analysis of mass spectrometry raw data, DEGseq1.38.3 software was utilized for differential gene expression analysis. DEGs were identified based on the criteria of |log2FoldChange| > 1 and *p*-value < 0.05. Visualization of DEGs between the 16F16-treated group and control group was achieved through a volcano plot. It was found that a total of 994 genes were differentially expressed after 16F16 treatment, with 330 genes up-regulated and 664 genes down-regulated, respectively (Figure 4A). The number of down-regulated genes was much larger than the number of up-regulated expressed genes. Therefore, it can be hypothesized that the inhibition of PDI activity after 16F16 treatment resulted in the affected folding of certain related proteins, which suppressed the trend of related gene expression. Notably, the top five genes that were significantly up-regulated were AT1G68250, AT2G36640, AT2G40220, AT1G54870, and AT1G63600. The top five genes that were significantly down-regulated were AT3G03670, AT3G48640, AT2G29220, AT2G13810 and AT1G21240 (Appendix A).

To further investigate the molecular mechanism of root shortening in Arabidopsis, changes in protein profiles between control and treated groups were analyzed using TMT comparative proteomics. Differentially expressed proteins (DEPs) screening criteria: fold change (ratio of the mean value of expression between the two groups) ≥1.5 or fold change ≤0.67 (this means ≤ 1/1.5) and *p* < 0.05. A total of 120 DEPs were identified, with 53 up-regulated and 67 down-regulated proteins. The top five up-regulated DEPs were Q8VYS0, Q9LY27, Q9SUI4, F4JQF1 and A0A178W2W5, while the top five down-regulated DEPs were A0A1P8AU94, A0A7G2EF89, A0A8F5Z876, O80517 and A0A178VRV6 (Appendix A). Three DEPs related to Arabidopsis root development were found, all of which were down-regulated (Appendix A).

Subcellular localization is a critical factor in determining protein function. Predictions for the subcellular localization of DEPs were conducted using the Uniprot website. According to the predicted results, DEPs were mainly located in the membrane (33, 40.24%) and cytosol (13, 15.85%). In addition, the localization of other DEPs in descending order is extracellular region, cell wall, nucleolus, vacuole, plastid, cytoplasm, chloroplast, cytoskeleton, photosystem II and endoplasmic reticulum (Figure 4C). Disulfide bonds play a crucial role in the folding of secreted proteins and plasma membrane proteins [4]. The fact that DEPs is predominantly localized to the cell membrane further supports previous findings. Thus, DEPs localized to the membrane respond to 16F16 treatment and might affect root length development in Arabidopsis seedlings.

### 2.5. Gene Ontology and Kyoto Encyclopedia of Genes and Genomes Pathway Analysis of Differentially Expressed Genes and Differentially Expressed Proteins

To investigate the regulatory mechanisms of the Arabidopsis root system following 16F16 treatment, the functional characteristics of DEGs and DEPs were classified through GO enrichment analysis. For RNA-seq, GO classification results showed that DEGs was significantly enriched in 10 terms in the biological process (BP), cellular component (CC), and molecular function (MF) categories (Figure 5). Specifically, the annotations of DEGs significantly enriched in the BP category were “response to stimulus”, “response to stress” and “response to chemical”, with the numbers of S genes (the number of significantly DEGs annotated in the designated database) were 344, 231 and 194, respectively. The high percentage of down-regulated genes in the biological processes mentioned suggests that 16F16 treatment may reduce plant response to stress by primarily inhibiting processes related to stimulus response, stress response, and chemical response. In the CC category, DEGs were significantly enriched in the “extracellular region”, “intrinsic component of membrane” and “integral component of membrane”, and integral component of membrane categories, with S gene numbers of 134, 319 and 303, respectively. In the MF category, the DEGs were significantly enriched in “tetrapyrrole binding”, “carbohydrate binding” and “heme binding”. The numbers of genes were 48, 42 and 43, respectively (Figure 5A).

KEGG pathway analysis was performed on the RNA-seq DEGs, and the top 20 significantly enriched were selected for analysis. DEGs were enriched in “phenylpropane biosynthesis” (n = 25) (Appendix A), “plant−pathogen interaction” (n = 13), “plant hormone signal transduction” (n = 16), “starch and sucrose metabolism” (n = 11), and “MAPK signaling pathway−plant” (n = 9) and other pathways were enriched. Except for the plant hormone signal transduction pathway, there were more down-regulated genes in the other four pathways (Figure 5B).

For proteome sequencing, the top 10 term of DEPs enrichment significance in BP, MF and CC branches were selected. In BP, DEPs are mainly involved in “reactive oxygen species metabolic process”, “hydrogen peroxide metabolic process”, “hydrogen peroxide catabolic process”, “response to oxidative stress” and “embryo development”. In CC, DEPs are mainly enriched in the “photosystem I reaction center”, “photosystem”, “photosystem I”, “photosynthetic membrane” and “thylakoid”. In MF, DEPs are mainly associated with “antioxidant activity”, “peroxidase activity”, “oxidoreductase activity, acting on peroxide as acceptor”, “tetrapyrrole binding” and “oxidoreductase activity” (Figure 5C).

The top 10 item significantly enriched in KEGG are “phenylpropanoid biosynthesis” (n = 7) (Appendix A), “photosynthesis” (n = 4), “biosynthesis of various plant secondary metabolites” (n = 3), “biosynthesis of secondary metabolites” (n = 13) and “metabolic pathways” (n = 19) (Figure 5D).

### 2.6. Analysis of the Association between the Proteomic and Transcriptomic Data

To reveal the correlations between DEGs and DEPs, we performed a combined analysis of the transcriptome and proteome. A total of 16 shared DEGs were detected. Fifteen DEGs showed the same expression trend, and one DEG showed an opposite expression trend. The genes detected in the proteome and transcriptome were divided into nine modules based on their expression patterns, and represented in a nine quadrant diagram (Figure 6A). In order to further investigate the biological pathways in which DEGs and DEPs play a role, a KEGG enrichment analysis was performed. The results of the combined analysis showed that DEGs and DEPs were co-enriched in 17 common pathways (Figure 6B), and were significantly enriched in the pathways of “phenylpropanoid biosynthesis”, “photosynthesis”, “starch and sucrose metabolism”, “plant−pathogen interaction”, “plant hormone signal transduction”, “pentose and glucuronate interconversions”, and “cysteine and methionine metabolism” (Figure 6C). DEGs and DEPs with the same expression trend were significantly enriched in six pathways: “phenylpropanoid biosynthesis”, “alpha−Linolenic acid metabolism”, “peroxisome”, “biosynthesis of secondary metabolites”, “plant−pathogen interaction” and “metabolic pathway”. KEGG enrichment analysis showed that the phenylpropanoid biosynthetic pathway was highly enriched by DEGs and DEPs, again demonstrating that the phenylpropanoid biosynthetic pathway is an important pathway affecting the root development of Arabidopsis seedlings under 16F16 treatment (Figure 6D).

### 2.7. Transcription Factor Analysis of 16F16-Treated Arabidopsis Seedlings

A significant number of transcription factors were activated following treatment with 16F16. Analysis of transcriptome data revealed 59 differentially expressed transcription factors, with basic helix–loop–helix (bHLH), MYB, MIKC_MADS, WRKY, and ERF being the most prominent. Among these, bHLH and MYB-type transcription factors exhibited the highest number of activated members, with nine each. MIKC_MADS, WRKY, ERF, NAC, LBD, GRAS, WOX, GRF, HD-ZIP-type transcription factors have 2–7 activated members. We focused on four transcription factors, including bHLH, MYB, MIKC_MADS, WRKY. For the bHLH group, there were four up-regulated and five down-regulated DEGs. For the MYB group, there were three up-regulated and six down-regulated DEGs. Similarly, the MIKC_MADS group had more down-regulated than up-regulated DEGs (Figure 7). And all seven DEGs in the WRKY group found to be down-regulated, suggesting a potential negative correlation with the expression of genes related to root development. The precise mechanism by which these transcription factors regulate genes related to root development remains unclear and requires additional validation.

## 3. Discussion

The root system is essential for plants to adapt to terrestrial life [30,31,32], with Arabidopsis serving as a model for studying plant development due to its well-defined root structure [33,34]. Previous research has shown that the PDI inhibitor 16F16 can effectively inhibit PDI activity in bovine and human models [25,27]. This study explores the inhibitory impact of 16F16 on AtPDIs (Figure 1). Subsequently, Arabidopsis seedlings were subjected to 6 days of darkness treatment with varying concentrations of 16F16. The results showed that root length of Arabidopsis seedlings decreased with increasing levels of the inhibitor 16F16. Complete inhibition of root growth was observed when added 5 μM 16F16 (Figure 2). Transcriptome analysis showed that DEGs were annotated in 68 differential pathways, of which the top three enriched pathways were “phenylpropane biosynthesis”, “plant hormone signal transduction” and “plant−pathogen interaction” (Figure 5B). Proteome analysis showed that DEPs were annotated in 23 differential pathways, of which the top three enriched pathways were “phenylpropanoid biosynthesis”, “photosynthesis” and “biosynthesis of various plant secondary metabolites” (Figure 5D).

Phenylpropanoids play a crucial role in rhizome development, with phenylpropanoid-based polymers like lignin providing mechanical support for plant growth and aiding in the transport of water and nutrients [35,36,37]. Recent studies have focused on regulating lignin content by inhibiting CCoAOMT activity [37]. COMT and CCoAOMT are known to control the synthesis of S-type and G-type lignin monomers, respectively [38,39]. In transcriptome analysis, we observed down-regulation of CCoAOMT expression in the phenylpropane synthesis pathway and screened to obtain 10 DEGs, all belonging to the peroxidase family (Appendix A). Additionally, the proteomic analysis revealed seven DEPs with down-regulated expression in the phenylpropane synthesis pathway, with two of them linked to root development. These proteins, which contain disulfide bonds in their structures, belong to the peroxidase family and involved in phenylpropane biosynthesis and are associated with lignin biosynthesis and degradation (Appendix A). Research has indicated that PDI plays a role in the formation and isomerization of disulfide bonds during protein folding [40]. Our data confirm that 16F16 stress leads to a reduction in AtPDI activity, impacting the proper formation of disulfide bonds in peroxidase synthesis. This disruption leads to a decrease in the expression of peroxidase family proteins within the phenylpropanoid biosynthesis pathway. This in turn affects lignin synthesis and degradation [41], ultimately leading to inhibition of Arabidopsis seedling root growth.

Auxin is essential for regulating a wide range of growth and developmental processes in plants, such as cell division, tissue differentiation, organ development, and physiological responses [42,43]. Throughout the entire lifespan of plants, auxin is involved in promoting the expression of early auxin-responsive genes such as AUX/IAA, Gretchen Hagen (GH3), and small growth hormone up-regulated RNA (SAUR) genes [31]. It has been observed that both inhibition and enhancement of auxin signaling can impact the growth of primary roots in seedlings [44]. In this study, 16 DEGs were found to be significantly enriched in the “plant hormone signal transduction” pathway, and six DEGs were identified in the auxin pathway. Interestingly, the expression of genes associated with GH3 and SAUR were up-regulated after 16F16 treatment (Appendix A). Previous research has shown that auxin-induced GH3 genes can mediate the inactivation of IAA by conjugation, which further attenuates auxin signal transduction [45]. In this study, up-regulated expression of the GH3 gene after 16F16 treatment enhanced regulation of IAA inactivation and further attenuated auxin signal transduction. These findings suggest that alterations in the auxin pathway, which is known to be crucial for root growth, may be responsible for the observed shorter root system.

PDI, a member of the thioredoxin (Trx) superfamily, plays a role in catalyzing protein disulfide bond formation and preventing the aggregation of misfolded proteins [46]. Treatment with 16F16 significantly inhibits PDI activity, impacting the redox balance in plants. The NADP-linked thioredoxin and glutathione systems are vital pathways for reducing agents in living organisms, with TRX and glutathione acting as key regulators of redox homeostasis [47]. These pathways also impact auxin transport and metabolism, thus establishing a connection between redox regulation and auxin signaling [48,49]. Previous research has demonstrated the involvement of glutathione and TRXs in the regulation of root meristem size, and the maintenance of root apical meristem by glutathione is connected to auxin signaling [41]. GSH is indispensable for the development of both primary and lateral roots, potentially through the down regulation of PIN-FORMED (PIN) proteins [49,50]. Analysis of DEGs revealed six glutathione-S-transferases (GSTs) that were responsive to 16F16 stress, with five of them belonging to the Tau family of GSTs (Appendix A). This emphasizes the distinct role of this gene subclass in the oxidative stress response and supports the findings of previous research [51]. Our findings provide a reference for further studies on the crosstalk between redox systems and auxin in root development in Arabidopsis.

During the young seedling stage of rootless tiller growth, starch and sucrose metabolism may be to accumulate energy for subsequent growth and promote growth conditions [52]. Transcriptome analysis showed that 11 DEGs significantly enriched in the “Starch and sucrose metabolism”. These findings support the notion that starch and sucrose metabolism play crucial roles in early development.

## 4. Materials and Methods

### 4.1. Plant Materials and Growth Conditions

Arabidopsis ecotype Columbia (Col-0) seed were used in this study. The Arabidopsis seeds were first vernalized at 4 °C for 2 days. Then, they were sterilized by surface treating with 75% alcohol three times for 1 min each, followed by sterilization with 20% sodium hypochlorite for 7–8 min. After that, the seeds were washed five times with sterile distilled water and sown in 1/2MS medium containing 16F16. Culture dishes containing seeds were placed vertically and treated in the dark for 6 days. After the 6-day period, the phenotype was photographed, and the changes in root length for each group were recorded and compared.

Treatment solution configuration method: 16F16 was configured into a stock solution at a concentration of 20 mM, which was then diluted with 1/2MS medium to form 1 μM, 2 μM, 3 μM, 4 μM and 5 μM working solutions, respectively.

### 4.2. Protein Disulfide Isomerase Activity Assay

Reductase activity assay for AtPDIs were performed by incubating 130 μM insulin, 0.5 μM AtPDI5 or AtPDI9, and varying concentrations of 16F16 (0.5 μM, 5 μM, and 50 μM), at 25 °C in 100 mM K-Pi buffer (containing 2.5 mM EDTA, 0.1 mM DTT, PH7.5). The aggregation of the B chains of insulin occurred when the A and B chains were reduced. The aggregation signal of the B chains was detected through light absorption at 650 nm, and this signal was used to characterize the reduction of insulin by AtPDI and its associated proteins. The reductase activity of AtPDI was determined by calculating the ratio of the maximum slope of the curve to the delay time.

### 4.3. RNA Extraction, RNA Sequencing, and Data Analysis

To investigate the transcriptional regulation mechanism of Arabidopsis under 16F16 stress, we performed Arabidopsis transcriptome sequencing using RNA-Seq. Each treatment consisted of three biological replicates. After 6 days of treatment, the plants from both the treatment and control groups were collected and frozen in liquid nitrogen. RNA was extracted from the samples using the OMEGA plant RNA extraction kit (Omega Bio-Tek, Norcross, GA, USA). The mRNA with a polyA structure in the total RNA was enriched using Oligo (dT) magnetic beads. Ion interruption was employed to break the RNA into fragments of approximately 300 bp in length. The first strand of cDNA was synthesized using a 6-base random primer and reverse transcriptase with RNA as a template, followed by synthesis of the second strand of cDNA using the first strand as a template.

After constructing the library, PCR amplification was used to enrich library fragments. Library selection was performed based on a fragment size of 450 bp. The library was then checked for quality using the Agilent 2100 Bioanalyzer. The total concentration and effective concentration of the library were measured. The libraries with different Index sequences were mixed in proportion to their effective concentration and the amount of data required. The mixed libraries were uniformly diluted to 2 nM and denatured using base denaturation to create single-stranded libraries. Following RNA extraction, purification, and library construction, the libraries underwent paired-end (PE) sequencing using Next-Generation Sequencing (NGS) on the Illumina HiSeq sequencing platform (Bioprofile Tech, Shanghai, China).

### 4.4. Transcription Data Filtering, Differential Gene Screening and Enrichment Analysis

Raw read sequences were obtained by removing joint contaminated sequences, as well as low quality sequences and sequences with an N ratio greater than 5%. The filtered sequences were then aligned to the reference genome using HISAT2 to localize them. Gene expression levels were estimated by counting the reads localized to genomic or exonic regions using Fragments per Kilobase per Million Mapped Fragments (FPKM).

In this study, we analyzed gene expression differentially using DESeq. The conditions for screening DEGs were as follows: expression differential multiplicity |log2FoldChange| > 1 and significance *p*-value < 0.05. We analyzed the concatenation of DEGs and samples from all comparison groups using the Pheatmap 1.0.12 software package in R language. The FPKM values of the DEGs in each sample were logarithmically transformed (base 2) before analysis. Two-way cluster analysis was performed based on the expression levels of the same gene in different samples and the expression patterns of different genes in the same sample. Distances were calculated using the Euclidean method, and hierarchical clustering was performed using the longest distance method (Complete Linkage).

The DEGs were counted and annotated using NCBI, Uniprot, GO and KEGG databases were used to annotate the DEGs and obtain detailed descriptions of the DEGs. The enrichment analysis of GO and KEGG identified the number of genes enriched and the significance of the *p* value. This analysis helped determine the main biological functions exercised by the DEGs.

### 4.5. TMT-Based Quantitative Proteomics and Data Analysis

#### 4.5.1. Sample Preparation

Arabidopsis tissues were suspended in 200 μL of lysis buffer (4% SDS, 100 mM DTT, 150 mM Tris-HCl pH 8.0) on ice. Broken tissue was stirred using a homogenizer and boiled for 5 min. The samples were further sonicated and boiled again for 5 min. Insolubilized cellular debris was removed by centrifugation at 16,000 rpm for 15 min. The supernatant was collected and quantified by BCA protein assay kit (Bio-Rad, Hercules, CA, USA).

#### 4.5.2. Protein Digestion

Protein digestion (200 μg per sample) was performed according to the FASP procedure described by Wisniewski, Zougman et al. Briefly, detergents, DTT, and other low molecular weight fractions were removed by centrifugation-facilitated duplicate ultrafiltration (Microcon unit, 30 kD) using 200 μL of UA buffer (8 M urea, 150 mM Tris-HCl pH 8.0). Then, 100 μL of 0.05 M iodoacetamide in UA buffer was added to block the reduced cysteine residues and the samples were incubated in the dark for 20 min. The filters were washed three times with 100 μL of UA buffer, followed by two washes with 100 μL of 25 mM in NH_4_HCO_3_. Finally, the protein suspension was digested with 4 μg of trypsin (Promega, Madison, WI, USA) in 40 μL of 25 mM in NH_4_HCO_3_ at 37 °C overnight and the resulting peptides were collected as filtrate. Peptide concentration was determined by nanodrop device using OD280.

#### 4.5.3. TMT Labeling of Peptides

Peptides in each sample were labeled with TMT reagent according to the manufacturer’s instructions (Thermo Fisher Scientific, Waltham, MA, USA). After drying equivalent combinations of labeled peptides, the multiply labeled samples were graded using the Pierce High PH Reversed Phase Peptide Grading Kit (Thermo Fisher Scientific). The peptide fraction from each grade was evaporated to dryness and stored at −80 °C for LC-MS analysis.

#### 4.5.4. LC–MS/MS Analysis

LC-MS/MS analysis was performed on a Q-activated mass spectrometer coupled to an Easy nLC (Thermo Fisher Scientific). Peptides from each grade were upsampled onto a C18 reversed-phase column (12 cm long, 75 μm inner diameter, 3 μm) in buffer A (2% acetonitrile and 0.1% formic acid) and separated using a linear gradient of buffer B (90% acetonitrile and 0.1% formic acid) at a flow rate of 300 nL/min over a 90 min period. The linear gradient was set as follows: 0–2 min, 2–5% Buffer B; 2–62 min, 5–20% Buffer B; 62–80 min, 20–35% Buffer B; 80–83 min, 35–90% Buffer B; 83–90 min, Buffer B maintained at 90%. MS data were collected using the data-dependent top15 method for HCD fragmentation by dynamically selecting the most abundant parent ions from survey scans (300–1800 *m*/*z*). Target values were determined based on predictive automatic gain control (pAGC). The target AGC value was 1 × 10^6^ for full mass spectra with a maximum injection time of 50 ms, and the target AGC value for MS2 was 1 × 10^5^ with a maximum injection time of 100 ms. The dynamic exclusion duration was 30 s. The resolution of the measurement scans was 70,000 at *m*/*z* 200, and was set to 35,000 at *m*/*z* 200. The normalized collision energy was 30. The instrument operates with peptide recognition mode enabled.

#### 4.5.5. Database Searches and Analysis

The final LC-MS/MS raw files were imported into the search engine Sequest HT in the Proteome Discoverer software (version 2.4, ThermoScientific) for database searching. The database used for the search was Uniprot-Arabidopsis 3702-136433-20230330.fasta, from the website (https://wwwuniprot.org/taxonomy/3702 protein database, accessed on 30 March 2022) and the main search parameters were set as shown in Appendix A.

#### 4.5.6. Bioinformatics Analysis

Bioinformatics data were analyzed using Perseus 1.5.5.3 software, Microsoft Excel 2010 and R statistical computing 4.0.5 software. We will screen for significantly differentially expressed proteins with a critical value of ratio fold change set at >1.5 or <0.67 with a *p*-value of <0.05. The proteins will then be grouped according to protein level by hierarchical clustering. To annotate the sequences, we extracted information from UniProtKB, Swiss-Prot, Kyoto Encyclopedia of Genes and Genomes (KEGG), and Gene Ontology (GO). GO and KEGG enrichment analyses were performed using Fisher’s exact test with FDR correction for multiple testing. GO terms will be classified into three categories: biological processes (BP), molecular functions (MF) and cellular components (CC). Enriched GO and KEGG pathways will be statistically significant at the *p* < 0.05 level.

### 4.6. Statistical Analysis

All experiments were repeated at least three times, and the test data were expressed as means ± SD. Statistical analysis was performed by Student’s *t*-test using SPSS 19.0 (SPSS Inc., Chicago, IL, USA) software. The significant difference and *p* value are illustrated in the figures or figure legends.

## 5. Conclusions

Our data reveals a clear positive correlation between the concentration of inhibitor 16F16 and the inhibition of AtPDI activity. Additionally, we observed a negative correlation between the concentration of 16F16 and the root growth of Arabidopsis seedlings. Specifically, there was almost no root growth observed when added 5 µM 16F16. Furthermore, transcriptomic analysis revealed that key genes associated with phenylpropane biosynthesis, plant hormone signal transduction, plant−pathogen interaction, and starch and sucrose metabolism were responsive to 16F16 treatment in Arabidopsis seedlings. Through proteome analysis and transcriptome and proteome conjoint analysis, we found that the phenylpropanoid biosynthetic pathway is an important pathway affecting the root development of Arabidopsis seedlings under 16F16 treatment. Several possible transcription factors related to Arabidopsis seedling root development, such as bHLH, MYB, MIKC_MADS, WRKY. The results of the present study increase our understanding of the mechanism Arabidopsis seedling root development under 16F16 treatment at the molecular level.

## Figures and Tables

**Figure 1 ijms-25-03596-f001:**
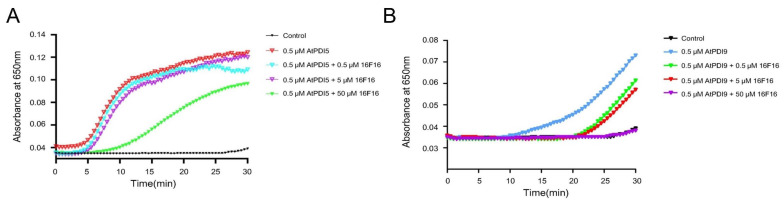
16F16 inhibits the activity of AtPDI in vitro. Reductase activity assay for AtPDIs were performed by incubating 130 μM insulin, 0.5 μM AtPDI5 or AtPDI9, and varying concentrations of 16F16 (0.5 μM, 5 μM, and 50 μM), at 25 °C in 100 mM K-Pi buffer. The aggregation signal of the B chains was detected through light absorption at 650 nm, and this signal was used to characterize the reduction of insulin by AtPDI and its associated proteins. (**A**) The reductase activity of AtPDI5 was determined under different concentrations of 16F16 treatment. (**B**) The reductase activity of AtPDI9 was determined under different concentrations of 16F16 treatment. The experiment was performed in triplicate and data are expressed as mean ± SEM.

**Figure 2 ijms-25-03596-f002:**
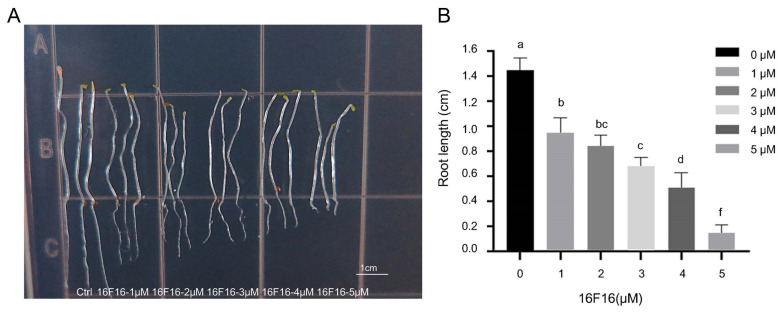
Effect of 16F16 stress on root elongation in Arabidopsis. 16F16 was configured into a stock solution at a concentration of 20 mM, which was then diluted into 1 μM, 2 μM, 3 μM, 4 μM and 5 μM working solutions with 1/2 MS medium, respectively. The phenotype was observed after 6 days of dark treatment. (**A**) Arabidopsis root phenotypes under 16F16 treatment. Scale bar, 1 cm. (**B**) Root length quantitative analysis of control and treated groups. The results were the means ± standard errors of 3 biological replicates (each of which has 3 technical replicate treatments, different treatment groups containing different lowercase letters indicate significant differences between treatments, *p* < 0.01, *p* < 0.001).

**Figure 3 ijms-25-03596-f003:**
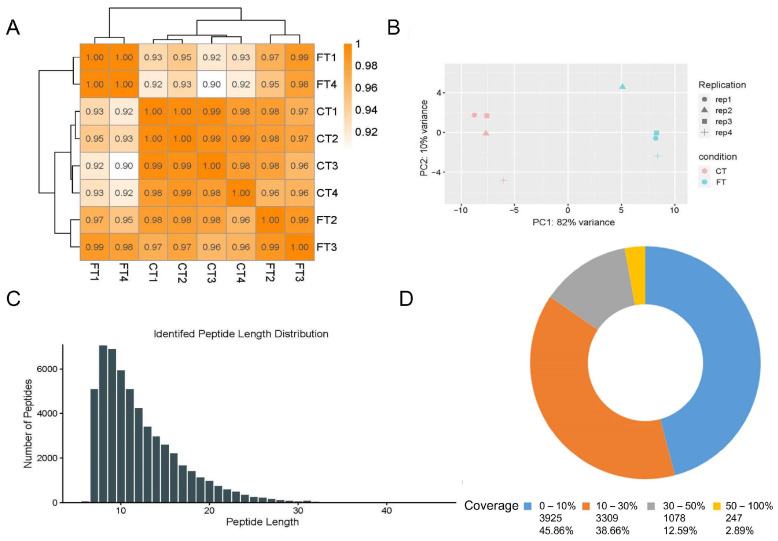
Transcriptome and proteome data quality assessment. (**A**) Sample correlation test. The Pearson correlation coefficient was used to indicate the correlation of gene expression levels between the samples, and the closer the correlation coefficient is to 1, the higher the similarity of expression patterns between the samples. (**B**) Transcriptome principal component analysis (PCA). The horizontal coordinate is the first principal component and the vertical coordinate is the second principal component. Different shapes in the graph indicate different samples and different colors indicate different groupings. (**C**) Length distribution of peptides defining each protein. (**D**) Sequence coverage range of the identified proteins.

**Figure 4 ijms-25-03596-f004:**
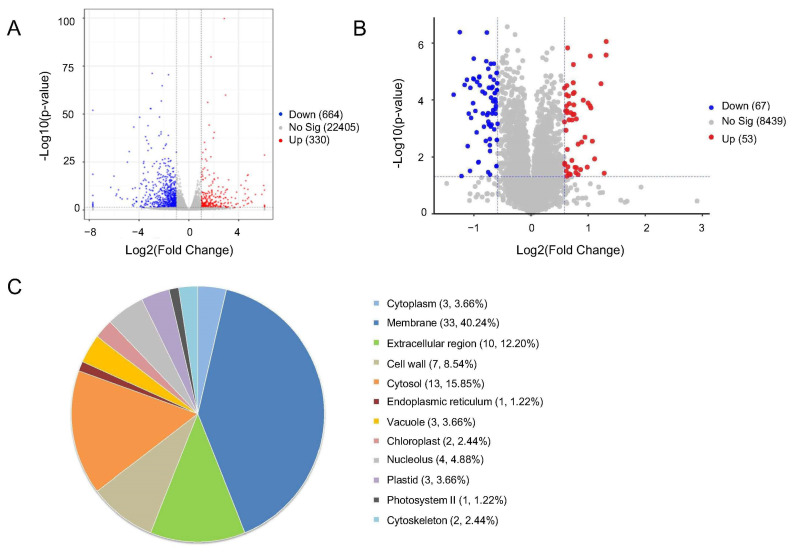
Analysis of DEGs and DEPs and subcellular localization of DEPs. (**A**) Volcano map of DEGs. Volcano plot of the number of DEGs identified by transcriptome analysis compared to WT. Each dot represents a gene. X-axis indicates log2 (Fold change) of DEGs and y-axis indicates −log10 (*p*-value). Red dots indicate up-regulated genes in the group, blue dots indicate down-regulated genes in the group, and gray dots indicate non-significant differentially expressed genes. (**B**) Volcano map of DEPs. Horizontal coordinates are log2 (Fold Change) and vertical coordinates are −log10 (*p*-value). Red dots signify up-regulated expressed proteins, blue dots indicate down-regulated expressed proteins, and gray dots represent non-significantly differentially expressed proteins within the group. (**C**) Subcellular localization prediction of the differentially expressed proteins.

**Figure 5 ijms-25-03596-f005:**
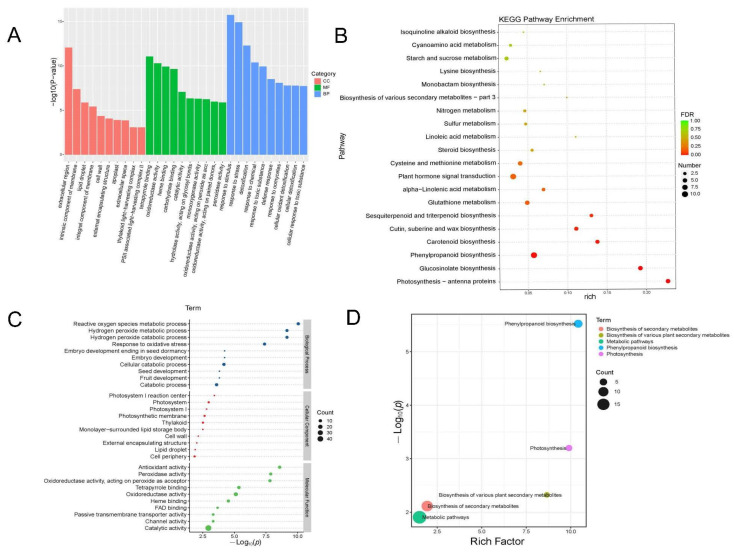
Gene ontology and Kyoto encyclopedia of genes and genomes pathway analysis of differentially expressed genes and differentially expressed proteins. (**A**) GO terms among DEGs. Horizontal coordinates are Go level2 ranked term, vertical coordinates are −log10 (*p*-value) enriched for each term. (**B**) KEGG enrichment analysis of DEGs. The horizontal coordinate is the pathway name and the vertical coordinate is the −log10 (*p*-value) enriched for each pathway. (**C**) GO classification and enrichment of DEPs. The horizontal coordinate is the negative logarithmic transformation of the enrichment significance *p*-value and the vertical coordinate is the GO term. Each circle denotes a term, and the circle size indicates the differential protein count. The 3 main branches are indicated by different colors. (**D**) KEGG enrichment for DEPs. Horizontal coordinates indicate enrichment factors and vertical coordinates indicate negative logarithmic transformations of *p*-values. Circle size indicates counts, and different colors indicate different paths.

**Figure 6 ijms-25-03596-f006:**
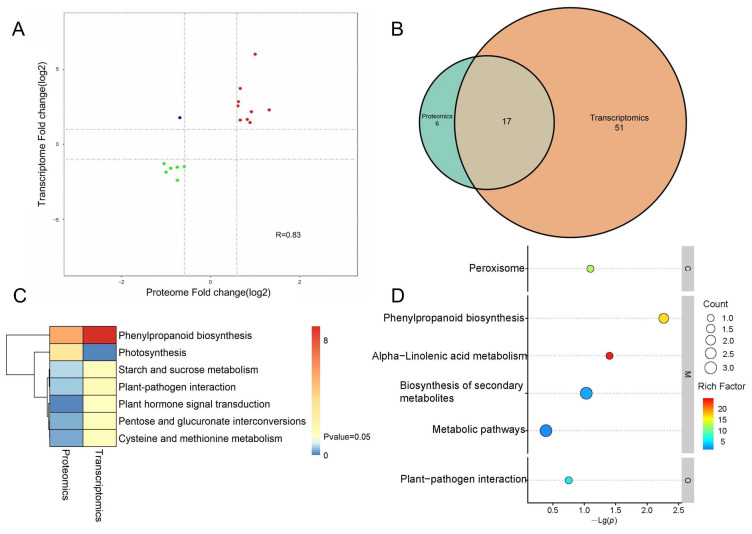
Combined analysis of DEGs and DEPs. (**A**) A nine-quadrant diagram of protein and mRNA associations. Each point represents one gene. Quadrants 1 and 9 indicate the genes negatively correlated with proteins. Quadrants 3 and 7 show the genes positively correlated with proteins. (**B**) Venn diagram illustrating the shared KEGG pathway in the treatment and control groups. (**C**) *p*-value heatmap. (**D**) KEGG enrichment analysis of DEGs with same expression trend as DEPs. Horizontal coordinate indicates the functional enrichment factor (rich factor), vertical coordinate indicates the negative logarithmic transformation of the *p*-value of the enrichment significance. Each circle in the figure indicates a pathway, which is denoted by different colors, and the size of the circle indicates the count.

**Figure 7 ijms-25-03596-f007:**
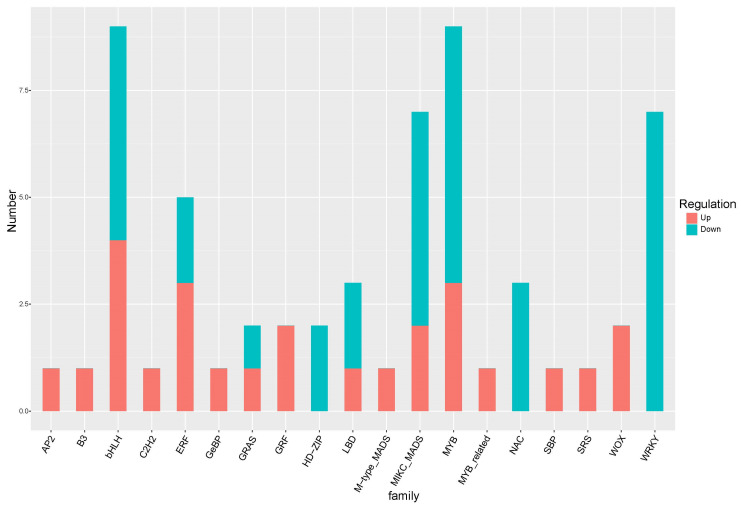
Transcription factor enrichment in 16F16 treatments. The x-axis and y-axis represent the transcription factors and their amounts in each classification, respectively.

**Table 1 ijms-25-03596-t001:** Statistic of RNA-seq data for each sample.

Sample	Reads No.	Clean Reads	Bases (bp)	Q30 (bp)	N (%)	Q20 (%)	Q30 (%)
CT1	43,871,862	40,883,152	6,624,651,162	6,239,513,563	0.000748	98.06	94.18
CT2	41,291,352	38,451,224	6,234,994,152	5,834,960,997	0.000696	97.81	93.58
CT3	42,679,328	39,797,694	6,444,578,528	6,069,764,697	0.000696	98.05	94.18
CT4	42,475,758	39,601,164	6,413,839,458	6,012,276,058	0.000709	97.87	93.73
FT1	40,937,568	38,143,410	6,181,572,768	5,792,475,690	0.000699	97.87	93.7
FT2	39,994,820	37,256,048	6,039,217,820	5,663,898,333	0.000709	97.89	93.78
FT3	43,689,960	40,734,992	6,597,183,960	6,179,973,447	0.0007	97.85	93.67
FT4	40,593,030	37,838,228	6,129,547,530	5,752,737,744	0.000697	97.92	93.85

Sample: the name of the sample; Reads No.: total number of Reads; Bases (bp): total number of bases; Q30 (bp): total number of bases with base recognition accuracy above 99.9%; N (%): percentage of ambiguous bases; Q20 (%): percentage of bases with a base recognition accuracy of 99% or more; Q30 (%): percentage of bases with base recognition accuracy of 99.9% or more.

## Data Availability

The data that support the findings of this study are available from the corresponding authors upon reasonable request.

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
