# Peer review of "Integrated Transcriptome and Proteome Analysis Reveals the Regulatory Mechanism of Root Growth by Protein Disulfide Isomerase in Arabidopsis"

_ijms, 2024, doi:10.3390/ijms25073596_

Round 1
Reviewer 1 Report
Comments and Suggestions for Authors
This manuscript reports a study on the potential regulatory mechanism of Arabidopsis root growth by Protein Disulfide Isomerase (PDI) based on expts, transcriptome and proteome analysis. To achieve their aim, 16F16, a small-molecule inhibitor of PDI was used. It is an interesting study and the results could provide new insights into the regulatory mechanisms of root growth. However, I think that the authors need to address following concerns before it can be published.
My major concern is that as shown in Figure 4, the authors only used a threshold of 1.2 to define significant changes in protein abundance is a fold change. Typically, a minimum fold change threshold for significant changes in protein abundance 1.5-fold or 2-fold. Is 1.2 too low? I think that the authors should re-analyse their data using a fold change of at least 1.5 as a threshold to make the analysed results more reliable and focused.
Other comments:
1. In the introduction, lines 36-38: author said ‘Protein disulfide isomerases (PDI) are thiol disulfide oxidoreductases that are primarily responsible for catalyzing oxidation, reduction and isomerization reactions of nascent secreted membrane proteins’ It is not fully accurate, as PDI can catalyse these reactions of both soluble and membrane proteins.
2. Lines 91-92: ‘16F16 is a small molecule compound with a 16 amino acid mutation that has the ability to inhibit protein disulfide isomerase’ It is unclear what ... a small molecule compound with a 16 amino acid mutation... means? More clear explanation about this inhibitor is required as it plays an important role in this study. Is it a specific inhibitor of PDI? Can it inhibit or affect the activity of other proteins?
3. Lines 108: ‘We chose PDI5 and PDI9 from Arabidopsis for our experiments...’ Why these two? Are there any reason for use PDI5 and PDI9?
4. Whilst PDI5 and PDI9 from Arabidopsis were used in this study, why hPDI is stated in the method?
5. What is the control in Figure 1?
6. Figure 2 and the method: what are the concentrations of 16F16 was added? Concentrations rather than volumes should be given, or at least state the concentration of the 16F16 stock solution.
7. The results of Figure5 are interesting and important, esp in terms of DEPs, but it is hard to read and thus understand. For example, the authors said: ‘The top 10 item significantly enriched in KEGG are “Glutathione metabolism”, “Bi-osynthesis of various plant secondary metabolites”, “Cyanoamino acid metabolism”, and “Tryptophan metabolism” (Figure 5D).’ However, it is unclear which proteins were identified and how they were affected (fold changes). It would be helpful to have a table to show the proteins identified in each processes.
8. In the sections 2.3-2.5, many abbreviations were used without a definition. Please define them first to help readers understand them
Comments on the Quality of English LanguageMostly fine and easy to read, but many abbreviations need to be define in the text fist before use.
Author Response
Thank you for your valuable suggestions, to which we have responded point by point, as detailed in the attached document.

Reviewer 2 Report
Comments and Suggestions for Authors
Dear Authors,
It was a pleasure to read your manuscript. There are a few changes in the text and a missing part of the conclusion and correlation between 16F16 stress and its differentially expressed gene information from transcritome sequencing.
Line 17: Change "16F16" to "16F16 (2-(2-Chloroacetyl)-2,3,4,9-tetrahydro-1-methyl-1H-pyrido[3,4-b]indole-1-carboxylic acid methyl ester)".
Line 21: Clarify "TMT" or provide its full form if applicable.
Lines 31, 33, 380: Ensure consistent italics for "Arabidopsis" and "A. thaliana" throughout the manuscript.
Line 36: Add "(-), i.e. thiol-disulfide oxidoreductase" after the term "PDI inhibitor".
Figure 1: Provide details for A) and B) in the figure legend.
Figure 2: A) Include a scale bar. Specify concentrations instead of uL for 16F16 in B). Remove unnecessary lines in the figure for clarity.
Line 315: Clarify whether "50uL" refers to concentration or volume.
Lines 367, 368, 369: Correct "AtGSTF2" to "AtGSTF2".
Line 385: Clarify the final concentrations of 16F16 in the volumes treated to Arabidopsis.
Additions:
Conclusion: Include a section summarizing the main findings and implications of the study.
Discussion:
Elaborate on the effects of 16F16 stress on Arabidopsis, particularly focusing on transcriptome data.
Correlate differentially expressed genes with the observed effects of 16F16 stress.
Discuss the implications of the transcriptome findings in relation to the aim of the study and the broader understanding of stress response mechanisms in plants.
By incorporating these revisions and additions, the manuscript will address the feedback provided and improve its clarity, consistency, and depth of discussion.
Author Response

(The authors gave the same response as above.)

Round 2
Reviewer 1 Report
Comments and Suggestions for Authors
The revised manuscript is much improved. However, I think that some further clarifications are required before it could be accepted for publication.
1. In Fig. 2B, please change X to the final concentration, such as 16F16 (mM).
2. Line 137: The description ‘… a fold change >1.5 or < 0.67’ is very confusing, please clarify it.
3. Figure 4 presentation needs to be better consistent with that described in the text lines 237-241. If a fold change>1.5 is used, why the plot presented differently? Also, not all data points consistent with the colour indications in the figure.
4. In the supplement tables, please add protein names in a column as well for easy reading and understanding.
Comments on the Quality of English Languagesee above
Author Response
Thank you for your valuable suggestions. We have carefully revised the issues raised, which are set in the attached document.

Reviewer 2 Report
Comments and Suggestions for Authors
Dear Authors,
It was a pleasure to read the revised manuscript entitled “Integrated Transcriptome and Proteome Analysis Reveals the Regulatory Mechanism of Root Growth by Protein Disulfide Isomerase in Arabidopsis”.
Thanks to all authors for considering my suggestions on this manuscript. The authors were going through all the comments and changes made to correct it on the revised manuscript. I am satisfied with the authors' responses. Therefore, I recommended this revised manuscript should be accepted in the current format and considered for the publication.
Thank you
Author Response
Thank you for your encouraging comments! Your help in revising and publishing the article is also appreciated.
